



# Using rapid damage observations from social media for Bayesian updating of hurricane vulnerability functions: A case study of Hurricane Dorian

Jens A. de Bruijn[1,2], James E. Daniell[1,3], Antonios Pomonis[1], Rashmin Gunasekera[1], Joshua Macabuag[1], Marleen C. de Ruiter[2], Siem Jan Koopman[4,5,6], Nadia Bloemendaal[2], Hans de Moel[2], Jeroen C.J.H. Aerts[2,7]

[1]World Bank Group 1818 H St NW, Washington DC, 20433, USA
[2]Institute for Environmental Studies IVM, VU University, Amsterdam, The Netherlands
[3]Geophysical Institute and Center for Disaster Management and Risk Reduction Technology, Karlsruhe Institute of Technology, Karlsruhe, Germany
[4]Department of Econometrics and Data Science, School of Business and Economics, Vrije Universiteit Amsterdam, The Netherlands
[5]CREATES, Aarhus University, Denmark
[6]Tinbergen Institute Amsterdam, The Netherlands
[7]Deltares, Delft, The Netherlands

*Correspondence to*: Jens A. de Bruijn (jens.de.bruijn@vu.nl)

**Abstract.** Rapid impact assessments immediately after disasters are crucial to enable rapid and effective mobilization of resources for response and recovery efforts. These assessments are often performed by analysing the three components of risk: hazard, exposure and vulnerability. Vulnerability curves are often constructed using historic insurance data or expert judgments, reducing their applicability for the characteristics of the specific hazard and building stock. Therefore, this paper outlines an approach to the creation of event-specific vulnerability curves, using Bayesian statistics (i.e., the zero-one inflated beta distribution) to update a pre-existing vulnerability curve (i.e., the prior) with observed impact data derived from social media. The approach is applied in a case study of Hurricane Dorian, which hit the Bahamas in September 2019. We analysed footage shot predominantly from unmanned aerial vehicles (UAVs) and other airborne vehicles posted on YouTube in the first 10 days after the disaster. Due to its Bayesian nature, the approach can be used regardless of the amount of data available as it balances the contribution of the prior and the observations.

## 1 Introduction

Natural hazards, such as tropical cyclones (Mendelsohn et al., 2012), floods (Winsemius et al., 2016) and earthquakes (Bilham, 2009), affect millions of people and cost billions of dollars in damages every year. Their impacts are expected to increase further (Cutter et al., 2015) often attributed to factors such as climate change (Mora et al., 2018) and increasing exposure (Bouwer, 2011). Damage assessment immediately after a natural disaster is crucial to enable rescue and relief organizations to mobilize resources for response and recovery processes (Kryvasheyeu et al., 2016). However, detailed and accurate assessments often take months to complete (UNDP, World Bank, 2013). Therefore, the World Bank has recently

employed the Global RApid post-disaster Damage Estimation (GRADE) approach to rapidly estimate damages to physical assets after major disasters (Gunasekera et al., 2018).

Damage estimations are commonly modelled using the three components of risk: hazard, exposure and vulnerability (Desai et al., 2015). Hazard is defined as a potentially damaging event, exposure as the elements subject to damage and losses as a

result of a hazard and vulnerability as "the conditions determined by physical, social, economic and environmental factors or processes which increase the susceptibility of an individual, a community, assets or systems to the impacts of hazards" (UNISDR, 2016). Vulnerability and fragility functions are commonly used to model damage to buildings due to natural disasters. These functions typically relate a measure of impact, such as wind speed (Pita et al., 2015), water depth (de Moel et al., 2013) or ground motion (Li et al., 2013), to damage. These relations are mostly based on previous insurance claims,

experiments or expert judgment and are often only applicable to particular hazard characteristics and specific built environments (Chung Yau et al., 2011; Douglas, 2007; Pita et al., 2013).

Observations, such as those from surveys (Wijayanti et al., 2017), can improve the accuracy of damage estimates (Douglas, 2007). However, affected areas are often difficult to access following a disaster (Bono and Gutiérrez, 2011), and human

resources are often limited (Koshimura et al., 2009). Data sources such as social media (Kryvasheyeu et al., 2016), unmanned aerial vehicles (UAVs) (Kim and Davidson, 2015) and other remote sensing techniques can provide detailed observations of damage quickly during and after a disaster. However, the amount of data is heavily dependent on the characteristics of the disaster area, such as the number of people that are able to use social media (de Bruijn et al., 2019; Yu et al., 2018) and cultural differences (Cho et al., 2009).


A scientific challenge is to seek for methods that use observations from the affected area to improve vulnerability curves. An example of such method is Bayesian analysis, which enables the updating of prior beliefs (e.g., beliefs based on expert judgment) with observational data, irrespective of the amount of data available (Koutsourelakis, 2010). The balance between prior information and observational data depends on the number of observations and the level of uncertainty in both the

observations and the prior beliefs.

Bayesian updating of fragility functions (i.e., the probability of exceeding a certain damage state) has been employed in numerous studies. For example, Li et al. (2013) combined results from numerical simulations and experimental testing of bridge substructures with Bayesian updating to obtain improved earthquake fragility functions. Mishra et al. (2017) updated

and quantified the uncertainty of analytical hurricane fragility functions for wood-frame buildings with experimental data. In another study, a Bayesian framework was designed to create fragility functions for earthquakes (Koutsourelakis, 2010). However, from a risk management perspective, vulnerability functions (where damage is represented as a damage ratio) are more desirable, as they measure the actual capacity of the built environment in an event (Rossetto et al., 2015).



In this paper, we aim to improve existing vulnerability functions employed in GRADE assessments by updating these functions with post-disaster observations using Bayesian zero-one inflated beta regression (Ospina and Ferrari, 2010). The method is applied to a case study of tropical cyclone wind damage in the Bahamas caused by Hurricane Dorian in September 2019. The observations were obtained from online media reports (i.e., YouTube) in the days following the disaster. This data includes some ground observations from driving cars but is mostly composed of observations from UAVs and videos shot

from other airborne vehicles. In the remainder of this paper, we describe the general methodology (Sect. 2) and its application to the Bahamas (Sect. 3). We also discuss the applicability of the proposed method to other disaster types and different data sources.

## 2 Methodology

Bayesian updating or inference refers to the process of updating existing knowledge for a set of n parameters Θ, defined as Θ

$= (\theta_1, \dots, \theta_n)$ and expressed in a prior distribution, defined as $P(\Theta)$, with new information X to find the posterior distribution, defined as $P(\Theta|X)$. Mathematically, we can express Bayesian updating as

$$P(\Theta|X) \propto P(X|\Theta) * P(\Theta). \qquad (1)$$

The posterior distribution $P(\Theta|X)$ is obtained by multiplying the prior distribution $P(\Theta)$ by the likelihood distribution $P(X|\Theta)$, which is the probability of observing X given parameter set Θ, and a normalizing constant. The normalizing constant

$P(X)$ is omitted here since this is automatically determined during the process of Gibbs sampling (see below; Gilks et al., 1996; Plummer, 2003b).

Following the Bayesian framework, the prior distribution $P(\Theta)$ for vulnerability must be defined. The vulnerability of the building stock can be expressed as a curve that maps wind speed as the explanatory variable $(v_1, \dots, v_n)$ to a damage ratio

from 0 to 1 (inclusive) as the response variable $(y_1, \dots, y_n)$. Fragility curves of individual components of buildings (e.g., roof sheathing and nails) are widely regarded as following a cumulative lognormal distribution (Ellingwood et al., 2004; Lee and Rosowsky, 2005; Li and Ellingwood, 2006). Assuming identical fragility curves for individual components and independence of failure, a vulnerability curve for a building follows that same cumulative lognormal distribution (Holmes, 1996):

$$y = \Phi\left(\frac{\ln(v/\alpha)}{\beta}\right) \qquad (2)$$

where $y \in (0,1)$ is the damage ratio, $\alpha$ the median capacity of the building stock, $\beta$ the logarithmic standard deviation of that capacity, $\Phi(\cdot)$ the cumulative probability density function for the standard normal distribution and $v$ the sustained wind speed. While $\alpha$ and $\beta$ can be expressed deterministically, we prefer to regard both parameters as uncertain and we express them as the random variables $\theta_1$ and $\theta_2$, respectively.






For proportional data, the beta distribution is often used as the basis for the likelihood function $P(X|\Theta)$ (Gupta and Nadarajah, 2004) because it supports a wide range of shapes on the interval (0, 1). Its probability density function, re-parameterized in terms of mean $\mu$ and precision $\varphi$, is given by Ferrari and Cribari-Neto (2004):

$$f_{beta}(y; \mu, \varphi) = \frac{\Gamma(\varphi)}{\Gamma(\mu\varphi)\Gamma((1-\mu)\varphi)} y^{\mu\varphi-1}(1-y)^{(1-\mu)\varphi-1}, \qquad y \in (0,1) \qquad (3)$$

where $\Gamma(\cdot)$ is the gamma function.

However, observations of the damage ratio can be true values of zero (i.e., no damage) and one (i.e., complete destruction), which are not supported by the beta distribution. In fact, such observations are more frequent because the assumption that the individual building components of buildings fail independently does not always hold. For example, houses are often

completely destroyed (i.e., damage ratio of 1) due to the collapse of an important fundament (Keote et al., 2015) or complete displacement of the entire building (Shultz et al., 2005). By contrast, houses may be completely undamaged due to their environment or protection measures, such as shielding by other standing buildings (Keote et al., 2015). Using the zero-one inflated beta distribution enables us to explicitly model these 0 and 1 observations through probabilities $\pi_0$ and $\pi_1$, respectively.


Therefore, for proportional response variables $y_i \in [0,1]$, Ospina and Ferrari (2010) propose a zero-one inflated beta regression. Here, the response variable $(y_1, \ldots, y_n)$ is modeled as a mixture probability function of $\pi_0$, the probability that $y = 0$; $\pi_1$, the conditional probability $Pr(y = 1 \mid y \neq 0)$; and the beta distribution with expected value $\mu_y$ and precision $\varphi$ for the values between 0 and 1 $(0 - 1)$. Its probability density function is given by:

$$f_{ZOIB}(y; \pi_0, \pi_1, \mu, \varphi) = \begin{cases} \pi_0 & if\ y = 0 \\ (1-\pi_0)\pi_1 & if\ y = 1 \\ (1-\pi_0)(1-\pi_1)f_{beta}(y; \mu_y, \varphi) & if\ y \in (0,1) \end{cases} \qquad (4)$$

By employing this distribution with proper parameterization (Sect. 3.3), we can model a process where it is highly probable that the damage ratio is 0 for low wind speed and 1 for high wind speed. For a wind speed in between, it is likely that $y$ is modelled on the continuous scale (0-1) through the beta distribution. We base the equation for $\mu_y$ on Eq. (2) presented above:

$$\mu_y = \Phi\left(\frac{\ln(v/\theta_1)}{\theta_2}\right) \qquad (5)$$

For probabilities $\pi_0$ and $\pi_1$ we specify a linear relationship with $v$ on the logit scale as follows:

$$\pi_0 = logit^{-1}(\theta_3 + \theta_4 v) \qquad (6)$$

$$\pi_1 = logit^{-1}(\theta_5 + \theta_6 v) \qquad (7)$$


In Eq. (5-7), the parameters $\theta_1, \dots, \theta_6$ are assumed to come from normal distributions with some mean and standard deviation.

Finally, the prior distribution (Eq. 4-7) and parameters $(\theta_1, \dots, \theta_6)$ can be updated with some observations with the wind speed as the explanatory variable $(v_1, \dots, v_n)$ and the corresponding response variables $(y_1, \dots, y_n)$ using Gibbs sampling, which is a Markov chain Monte Carlo (MCMC) algorithm, see Gilks *et al.*, (1996), for a general treatment. For the purpose of Gibbs sampling, the predictor variable $v$ is normalized such that $\tilde{v} \in (0,1]$ by dividing $v$ by the maximum observed wind speed (i.e., $\max(v)$). All other input variables are scaled accordingly. Then, samples of the posterior distribution are generated using the Just Another Gibbs Sampler program (JAGS; Plummer, 2003a). We first use 1,000 iterations to tune the samplers (i.e., adaptation), 1,000 iterations as a burn-in to find the place where the Markov chain is most representative of the sampled distribution, followed by 100,000 iterations in three chains with a thinning of 100.

To verify the convergence of the Markov chains, we can present different diagnostics which are reviewed in, amongst others, Cowles and Carlin (1996) and Brooks and Roberts (1998). In particular, we concentrate on diagnostics based on distributional and autocorrelation statistics.

## 3 Case study of Hurricane Dorian

In this section, the methodology for updating vulnerability curves described above is applied to a rapid damage estimation of Hurricane Dorian in Grand Bahama and the Abaco Islands, the northernmost main islands of the Bahamas. First, the hazard (Sect. 3.1) and exposure components (Sect. 3.2) are briefly described. Then, we discuss the collection of observations for the vulnerability component and the Bayesian updating process (Sect. 3.3). Finally, these three components of risk are combined to obtain a final damage estimate (Sect. 3.4). It should be noted that all data sources had to have been collectible in the first 10 days following the first landfall of the Hurricane in the Bahamas to be eligible for the rapid damage estimate.

### 3.1 Hazard

On 24 August 2019, Tropical Storm Dorian formed over the Atlantic Ocean and began tracking through the Windward Islands and the U.S. Virgin Islands towards the Bahamas (Avila et al., 2020). On 1 September, Dorian reached Category-5 strength with maximum sustained wind speeds exceeding 300 km/h. At 16:40 UTC that day, Dorian made landfall at Elbow Cay, Great Abaco, in the Bahamas. During its passage over the Great Abaco and Grand Bahama islands, the weakening of the nearby high-pressure area caused the hurricane to lose its steering currents and therefore significantly slowing its forward speed to 2 km/h, at times even coming to a standstill. Dorian remained near-stationary for 36 hours, until the hurricane started moving north-northwestwards towards North Carolina (USA). Dorian dissipated on 8 September near Canada.





160    In this study, meteorological conditions during Hurricane Dorian's passage over the Bahamas is taken from the International
Best Track Archive for Climate Stewardship (IBTrACS; Knapp et al., 2010). Using the position of the eye, the minimum
pressure, maximum wind speed, and size of the eye (i.e., radius to maximum winds) the 2D wind field is constructed by
applying the parametric approach of Holland (1980), which was further refined in by Lin and Chavas (2012). To obtain the
wind field at 10 meter above surface level (Fig. 1) a reduction factor of 0.85 is applied (Powell et al., 2005).

165    **3.2 Exposure**

To determine the exposure of the residential buildings on the Abaco and Grand Bahama Islands, we consulted the 2000 and
2010 Population and Housing Census of the Bahamas (Department of Statistics, 2002, 2012). The census contains
information about the housing stock. Specifically, it contains data on residential buildings and occupied and vacant dwelling
units for each settlement and enumeration district in the Abaco Islands and each supervisory district in Grand Bahama, as
well as the number of bedrooms and the total annual household income for each household size. In the Abaco Islands (and to
a lesser extent in Grand Bahama), the proportion of vacant housing stock is relatively high as the islands are a tourist
destination, and many homes are owned or rented by vacationers. The Abaco Islands also has many migrant communities
(mainly working in the service sectors for the tourism industry), who reside in low-quality housing in several informal
settlements, such as the Mudd and the Pigeon Peas settlements. Other settlements are occupied by the local Bahamian
population, while tourists and foreign citizens, often reside in high-value homes and resorts.

To account for this heterogeneity all settlements and supervisory districts ("regions") were mapped individually and the
value of residential buildings within three building classes (low-, medium- and high-quality) was estimated for each region.
We first estimated the number and area of buildings per building type in each region. To do so, we consulted building
footprints from OpenStreetMap (and found 16,100 and 12,500 building footprints in Grand Bahama and the Abaco Islands
respectively). We estimated the housing stock in 2019 by projecting data from the 2000 and 2010 census onto 2019, taking
official population projections into account (Department of Statistics, 2015) and estimated the floor area of the dwellings
using data on the number of bedrooms per dwelling. We also consulted the Bahamas 2013 Household Expenditure Survey
(Department of Statistics, 2016), which provided information about household consumption quintiles and housing
conditions, such as type of dwelling, number of rooms, bedrooms, period of construction of the dwelling, type of tenure, type
of construction material used for the outer walls, the roof cover and the floors.

Next, we estimated the unit cost of construction of the building classes based on housing prices provided by real estate
agents, building contractors and a census of informal settlements in the Abaco Islands (Shanty Town Task Force, 2018).
Finally, we consulted the Annual Building Construction Statistics Reports (e.g., Department of Statistics, 2018), which
contain data on the investment values of newly constructed buildings. Figure 2 shows the reported monetary value of low-
quality residential buildings per region (Fig. A1 and A2 for values of medium- and high-quality buildings; note the adjusted





scales). Note that these estimations do not include building content. Since no official maps of the regions were available, we determined the coordinates of the population centres for each region. All maps are presented as Voronoi diagrams (i.e.,

partitioned into regions closest to each centre point) based on these centroids.

### 3.3 Vulnerability

To derive event-specific vulnerability curves, we aimed to update vulnerability curves derived from previous hurricane observations in similar built environments using damage observations for individual buildings in the affected area. Therefore, we analysed all 498 YouTube videos that were listed when we searched for "*Bahamas Dorian*" and that were

posted in the 10 days after the first landfall of Hurricane Dorian in the Bahamas (September 1st–9th). We then analysed all videos that 1) showed an overview of an area or a row of buildings (to ensure the sample was as representative as possible), 2) that we were able to geotag (i.e., locate) and 3) that showed buildings that did not appear to have undergone extensive flood damage. This resulted in a set of 15 videos (Appendix B), from which we extracted 732 buildings. Figure 3 depicts two examples of buildings extracted from the videos. By comparing the footage with satellite imagery of the area before the

hurricane, we ensured that all buildings in an observed area or row of buildings were analysed, including those that completely disappeared in the storm.

Then, the damage ratio [0-1] and building class (i.e., low-, medium, and high-quality) were estimated for each building. Based on experience in post-disaster damage assessment in insurance, economic damage ratios were estimated based on the

damage seen in the value of subcomponents of the structure and their relative values and interactions as a whole, see Massarra et al. (2019). In some areas, especially those with low-quality houses, it was difficult to extract an image of each individual building due to the large amount of destruction and displacement. In such cases, we estimated the number of buildings per level of damage from pre-event satellite imagery.

Next, we derived a prior (i.e., a vulnerability curve based on pre-existing knowledge; Sect. 2) for each building class by estimating the parameters based on expert judgment of the strength of the buildings in similar built environments. Such curves have fitted the results of previous PDNAs within the Caribbean for stronger wind events for economic damage, and provide similar smoothed curves to existing models in the region such as those presented in the UNDRR's Global Assessment Report on Disaster Risk Reduction (UNISDR, 2015) for developed locations. Figure 4 (left) shows the

parameters $\alpha$ and $\beta$ of these curves for the three housing qualities and Fig. 4 (right) shows the associated damage ratio for low-quality buildings in the affected region.

We then set the priors (Fig. 5) for $\mu_{\theta_1}$ and $\mu_{\theta_2}$ (Eq. 5) using the specified values of $\alpha$ and $\beta$ for each building class; $\sigma_{\theta_1}$ was set to 15 using the uncertainty range expressed in the fragility curves for a specific damage state (i.e., the range of wind gust

speed that causes a specific damage state) in HAZUS (a tool for analysing natural hazards in the United States) for a similar


built environment (Vickery et al., 2006) and $\sigma_{\theta_2}$ was set to 0.03 to allow for uncertainty of the building typology. The values of the means and standard deviations of $\theta_3, \ldots, \theta_6$ (Eq. 6, 7) were set such that the probability $\pi_0$ is near one when $\mu_y$ is near zero and $\pi_0$ near zero when $\mu_y$ is near one. Conversely, $\pi_1$ is set such that its value is near one when $\mu_y$ is near one and $\pi_1$ near zero when $\mu_y$ is near zero. This is explained in full in Appendix C. For precision parameter $\varphi$, we use an uninformed

uniform prior.

For each building's observed damage ratio $(y_1, \ldots, y_n) \in [0,1]$, the maximum sustained wind speed $(v_1, \ldots, v_n)$ was obtained using the location of the building (Sect. 3.1). Finally, using these observations we performed Gibbs sampling to obtain the posterior distribution (Fig. 5; Sect. 2). In Fig. 5, the upper row (a-c) shows the prior vulnerability curves, whereas

the lower row (d-f) shows the posterior curves. From left to right the columns represent the vulnerability curves for low- (a,d), medium- (b,e) and high-quality (c,f) buildings. The red, blue and green colours denote respectively $\pi_0$, $\pi_1$ and $\mu_y$. The solid line represents the median ($P_{50}$), the dashed lines ($P_{25-75}$) represent the 25th and 75th percentile and the shaded area represents the 10th-90th percentile range.

Comparing the top and the bottom row, the curves for all building types have shifted substantially showing the result of the Bayesian updating. Note that for the posterior vulnerability curves for low-quality buildings (bottom left; d) it appears that there is a significant probability that the building is entirely destroyed (green curve). However, this is not the case since $\pi_1$ is the conditional probability that the damage ratio is one given that the damage ratio is not zero ($Pr(y = 1 \mid y \neq 0)$; Eq. 4).

Figures D1, D2 and D3 display the values per iteration, density plot and autocorrelation for $\theta_1, \ldots, \theta_6$ for each building quality class. While most sequences of generated parameters in the MCMC have low autocorrelations, some parameters do show high autocorrelations (i.e., $\theta_3$ and $\theta_4$ for low- and medium- quality buildings and $\theta_1$ and $\theta_2$). This is likely caused by the absence of data information for these parameters, together with a possibly large disagreement between the prior and observed data.

Figure 6 shows the posterior damage ratio per district for low-, medium- and high-quality buildings.

**3.4 Damage estimation**

Finally, the three components of risk were combined (i.e., hazard × exposure × vulnerability) to obtain a damage estimation for residential buildings (Table 1; Fig. 7) of 1056 million USD using the prior vulnerability curves versus an estimate of 658

million USD using posterior curves (i.e., ~38% lower). It should be noted that we used the coordinates of the population centre determined for each region (Sect. 3.2) to extract the wind speed data (Sect. 3.1).



## 4 Discussion and conclusions

In this paper, we present a framework that uses Bayesian updating with social media data (YouTube videos) to create event-specific vulnerability curves. This framework uses the zero-one inflated beta distribution, which allows us to use post-disaster observations to create vulnerability curves that have been adjusted for local hazard and building characteristics. We demonstrate their application in a rapid damage assessment of structural damage to buildings caused by Hurricane Dorian in the Bahamas. In our estimation, wind damage to residential buildings is ~38% lower compared to that calculated using pre-existing vulnerability curves (i.e., the priors). The largest relative differences were found for medium- and high-quality buildings, which we argue are most likely designed to be designed according to strict building codes (Ministry of Works & Utilities, 2003).

However, using social media data to assess building damages has several limitations. Observations from online media are biased, and some demographic groups have easier access to internet resources than other groups (Duggan and Brenner, 2013). In addition, observations tend to focus on the most impacted areas, as these are more newsworthy (Miles and Morse, 2007). While we have aimed to reduce this bias by only including footage that showed relatively large areas, we found very little footage of the less severely impacted parts of the islands, such as the northern and southern tips of the Abaco Islands. Moreover, the large spread in the observations (Fig. 5) shows that vulnerability is a complex concept. The vulnerability of a single building or part of building to a specific hazard is determined by many factors. In this paper, we only considered sustained wind speed. However, rainfall patterns and other environmental factors could also be important (Hatzikyriakou et al., 2016; Knapp et al., 2010). In addition, while we used three different building classes, the real variation in the strength of these buildings cannot be captured by three relatively simple curves.

The variation in building classes was also difficult to capture. For our case study, we based our classification of damaged buildings on post-disaster imagery. While we aimed to deduce the building quality class from the building structure rather than from the damage to the building, it is likely that bias was introduced in the classification. A better approach would be to use pre-disaster imagery (e.g., Google Street View or Mapillary) or, even better, detailed construction data per building. Unfortunately, these were unavailable for the Grand Bahamas and the Abaco Islands.

This was further complicated by the limited availability of vulnerability curves for the specific built environment. The estimated priors are based on a combination of data sources and expert judgement. This likely caused a relatively large disagreement between priors and observations for some parameters, resulting in a high autocorrelation (Fig. D1, D2 and D3). This calls for the establishment of a database of vulnerability curves for hurricane winds that considers a wide range of building typologies and strengths across a wide spectrum of wind intensities.



We applied the method using observations from online media. However, point observations from other sources could also be used. For example, survey data collected by experts or insurance claims would likely be more reliable. However, the availability of such observations from such sources during the first days following a disaster is generally limited, which reduces their applicability for rapid post-disaster damage assessments. Our method could be applied to other hazard types, such as floods and earthquakes. However, building damage caused by standing water may be more difficult to observe from

pictures than structural damage caused by earthquakes. Therefore, for floods additional data, such as data from surveys or insurance reports, may be required.

## 5 Appendices

### Appendix A

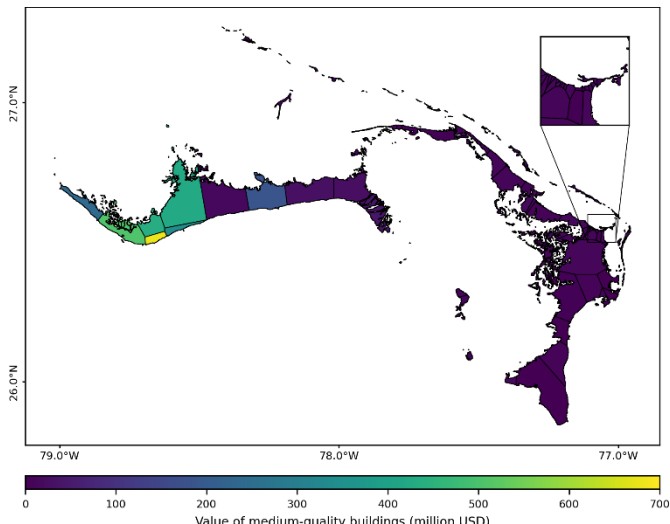

**Figure A1: Value of medium-quality buildings per region.**


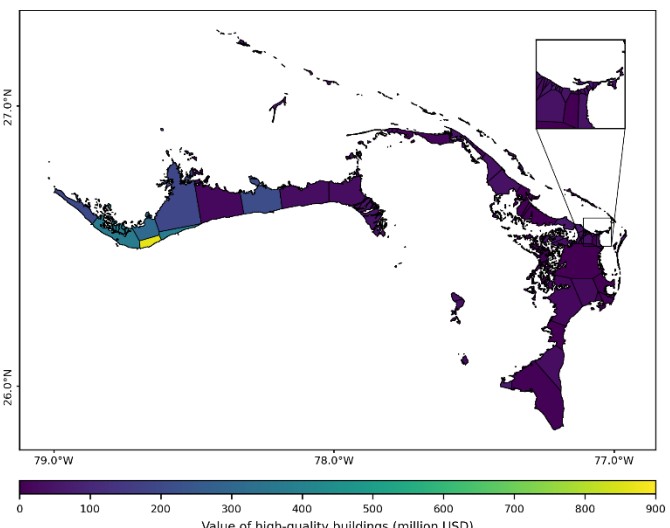

**Figure A2: Value of high-quality buildings per region.**

## Appendix B

- https://www.youtube.com/watch?v=rq95eJWxpd8
- https://www.youtube.com/watch?v=beXg9egFcAs
- https://www.youtube.com/watch?v=U0omTvsIr_U
- https://www.youtube.com/watch?v=SuzXtIdZqvg
- https://www.youtube.com/watch?v=8QoqtB6HPMY
- https://www.youtube.com/watch?v=SN4jgJX0OP8
- https://www.youtube.com/watch?v=hvCQtLWW-y4
- https://www.youtube.com/watch?v=11lZzCpeILs
- https://www.youtube.com/watch?v=SrUfwnX-UjI
- https://www.youtube.com/watch?v=Ar2BdIEI59w
- https://www.youtube.com/watch?v=9PypGi8M29Q
- https://www.youtube.com/watch?v=mpgGw3iyBPU
- https://www.youtube.com/watch?v=AdHombKmG78
- https://www.youtube.com/watch?v=_tFfnIGq2qE
- https://www.youtube.com/watch?v=E-zdt-fYLlg

## Appendix C

$\mu_{\theta_3}$ and $\mu_{\theta_4}$ are chosen such that $\pi_0$ (i.e., the probability that the damage ratio is zero) is 0.99 where $\mu_y$ is 0.01 and $\pi_0$ is 0.01 where $\mu_y$ is 0.05. In simpler terms this means that the probability that the response variable $y_i$ is a true zero, is near-one when $\mu_y$ is also near-zero, and near-zero otherwise ($P_{50}$ or median in Fig. 5). The standard deviation is set at 10% of the mean. Likewise, $\mu_{\theta_5}$ and $\mu_{\theta_6}$ are chosen such that $\pi_1$ (i.e., the probability that the damage ratio is one) is 0.01 where $\mu_y$ is 0.95 and $\pi_1$ is 0.99 where $\mu_y$ is 0.99. Likewise, the standard deviation is set at 10% of the mean.




**Appendix D**

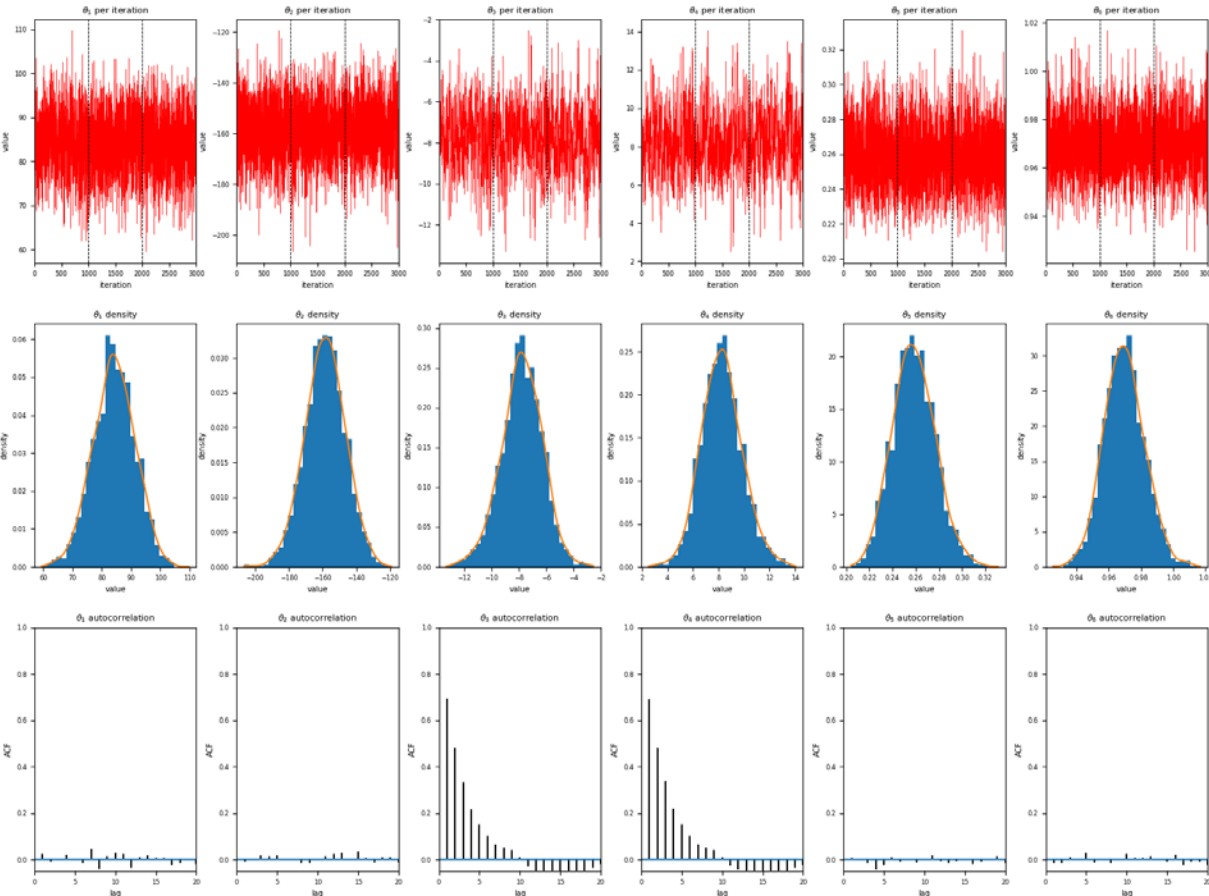

**Figure D1: Values per iteration, density plot and autocorrelation of $\theta_1, \ldots, \theta_6$ for low-quality buildings. The dashed lines in the top row represent the cut-offs between the 3 chains used in Gibbs sampling.**





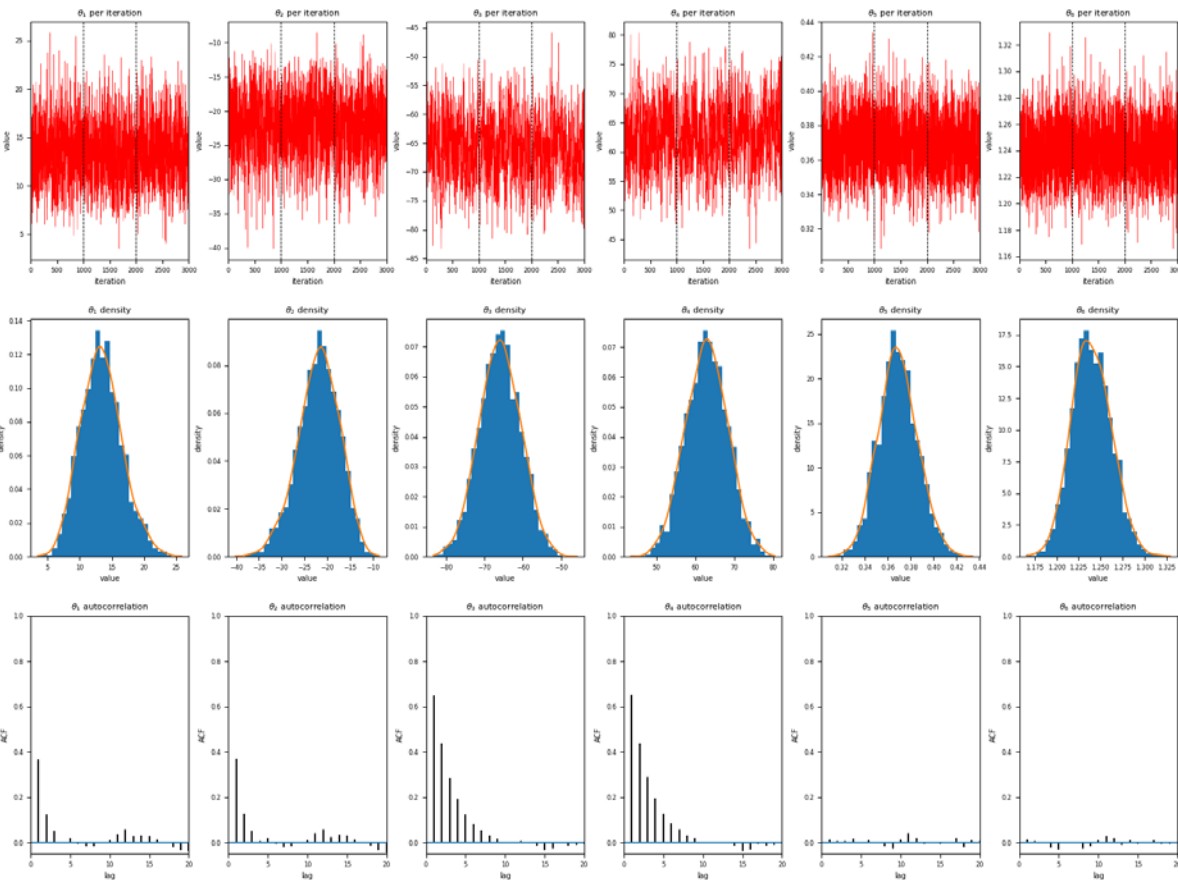

Figure D2: Values per iteration, density plot and autocorrelation of $\theta_1, ..., \theta_6$ for medium-quality buildings. The dashed lines in the top row represent the cut-offs between the 3 chains used in Gibbs sampling.



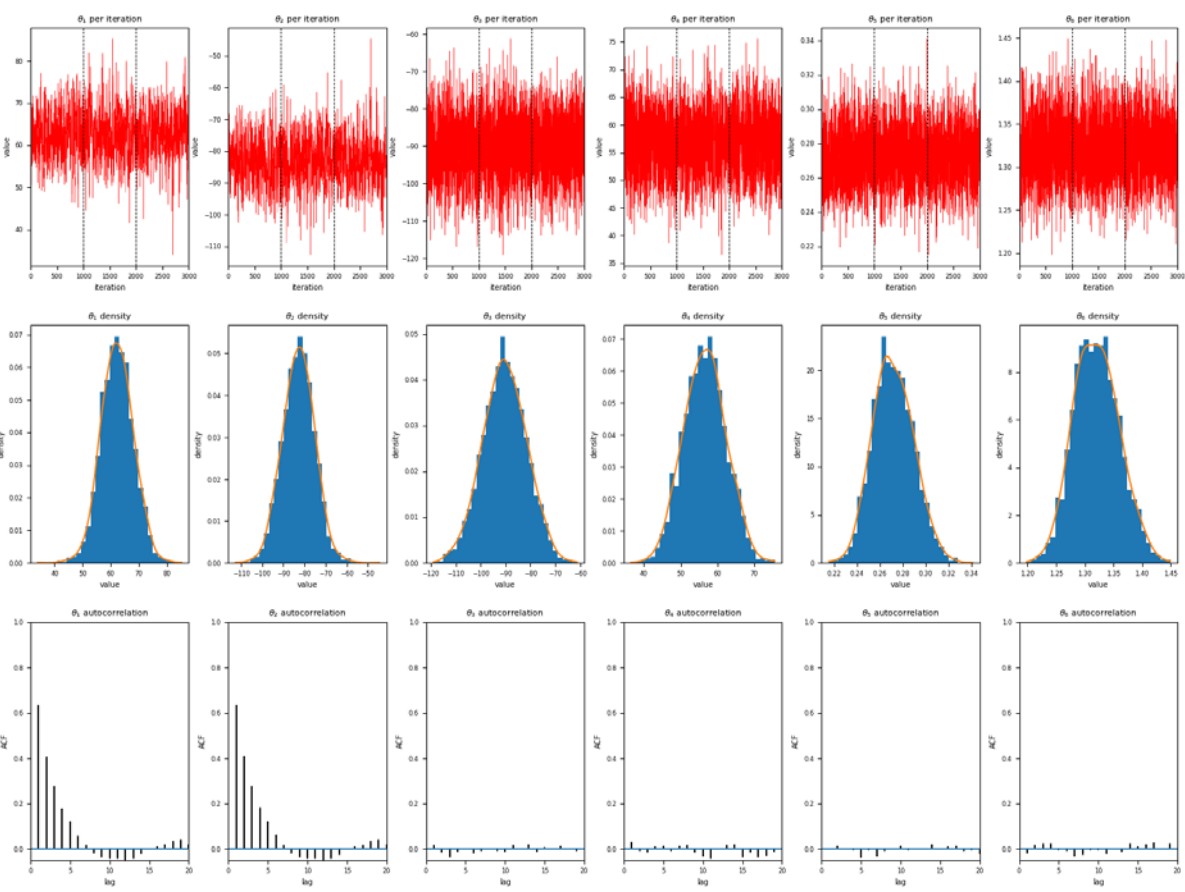

**Figure D3: Values per iteration, density plot and autocorrelation of $\theta_1, \dots, \theta_6$ for high-quality buildings. The dashed lines in the top row represent the cut-offs between the 3 chains used in Gibbs sampling.**

**6 Code availability**

Python and R code is available on GitHub (https://github.com/jensdebruijn/Bayesian-updating-of-hurricane-vulnerability-functions).

## 7 Data availability

The damage observations and wind field are available on GitHub (https://github.com/jensdebruijn/Bayesian-updating-of-
hurricane-vulnerability-functions).

## 8 Author contribution

JB developed the methodology and took lead in writing the manuscript. JD took part in the creative process and assessed the
damage ratios. AP, RG and JM collected and analyzed the exposure data. MR assisted in writing the manuscript. SK and HM
assisted in development of the methodology. NB created the wind field. JA assisted in writing and oversaw the creative
process.

## 9 Competing interests

The authors declare that they have no known competing financial interests or personal relationships that could have appeared
to influence the work reported in this paper.

## 10 Disclaimer

The findings, interpretations, and conclusions expressed in this paper are entirely those of the authors. They do not
necessarily represent the views of the International Bank for Reconstruction and Development/World Bank and its affiliated
organizations, or those of its Executive Directors or the governments they represent.

## 11 Acknowledgements

Our research was funded by an NWO-Vici grant from the Netherlands Organisation for Scientific Research (NWO; grant
number 453-14-006) and an EU-ENHANCE grant from the European Community's Seventh Framework Programme (FP7;
grant number 308438). This work was conducted by the World Bank's Disaster Resilience Analytics and Solutions (D-RAS)
Knowledge Silo Breaker (KSB), under the Global Practice for Urban, Disaster Risk Management, Resilience and Land
(GPURL). This research has been funded by the World Bank and the Global Facility for Disaster Reduction and Recovery
(GFDRR) grant (TF0B1140).

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

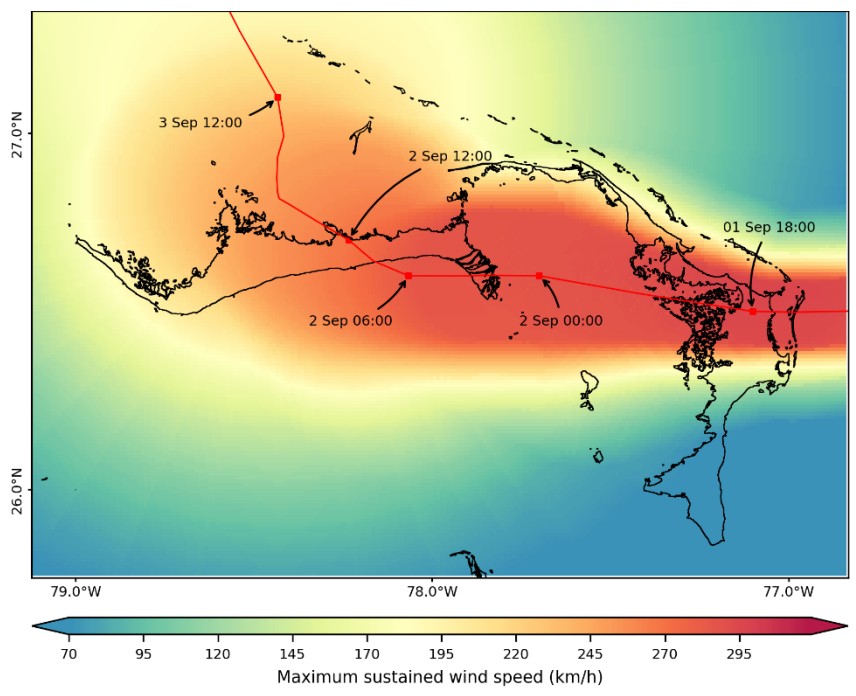


**Figure 1: Maximum 1-minute average sustained wind speeds at 10 meter above surface level at 0.01° resolution for Hurricane Dorian during its passage over the Bahamas in September 2019. Time stamps are given in UTC.**


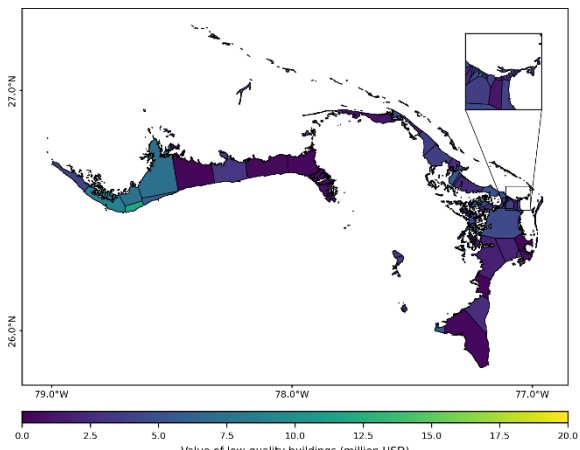

**Figure 2: Value of low-quality buildings per region.**

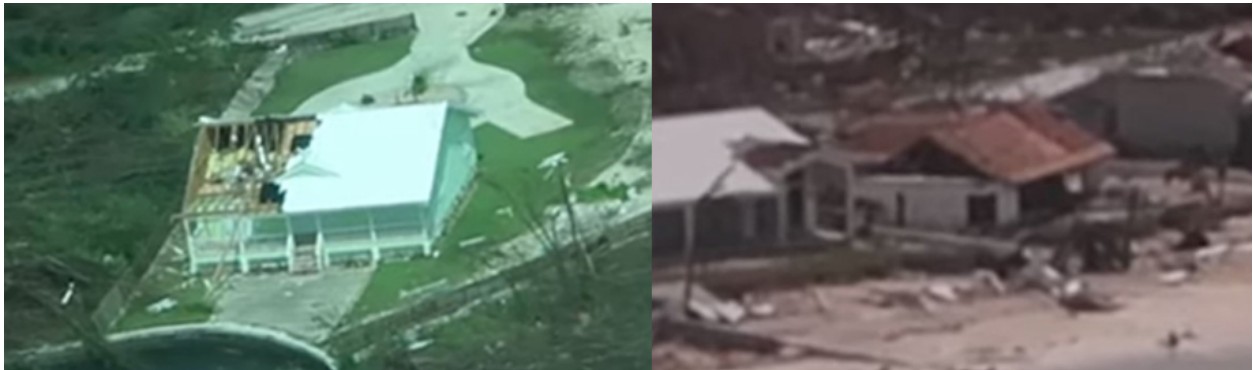


**Figure 3: Footage from "Dorian's Destruction of Treasure Cay, Abaco Bahamas" (https://www.youtube.com/watch?v=hvCQtLWW-y4) containing (left) a medium quality building with structural detailing with roof damage and water intrusion to part of the building's structure, non-structural damage (ca. 25% damage) and (right) a low quality building (with minor structural detailing), missing its front wall, roof damage, and significant debris (ca. 50% damage).**

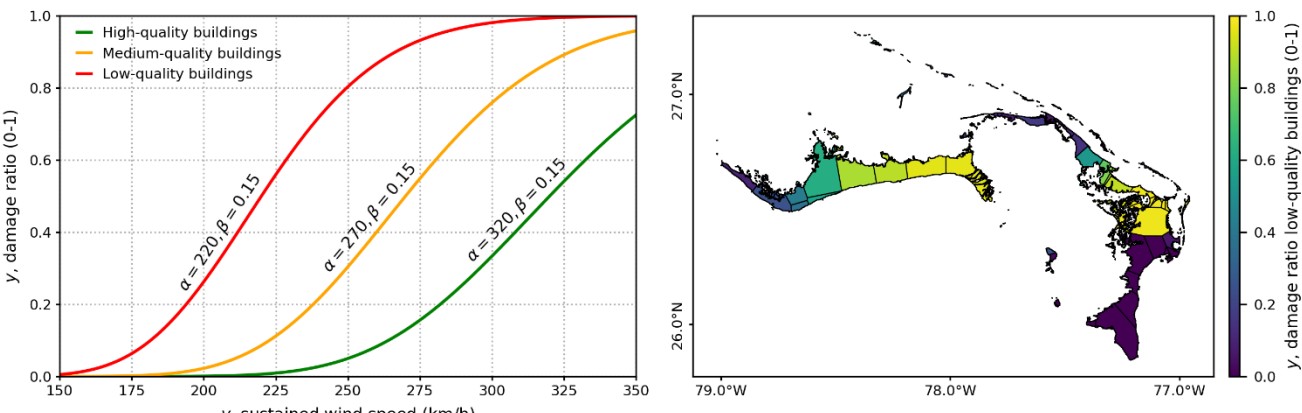


**Figure 4: Vulnerability functions for low-, medium- and high-quality buildings (left) and damage ratios for low-quality buildings in Grand Bahama and the Abaco Islands (right).**


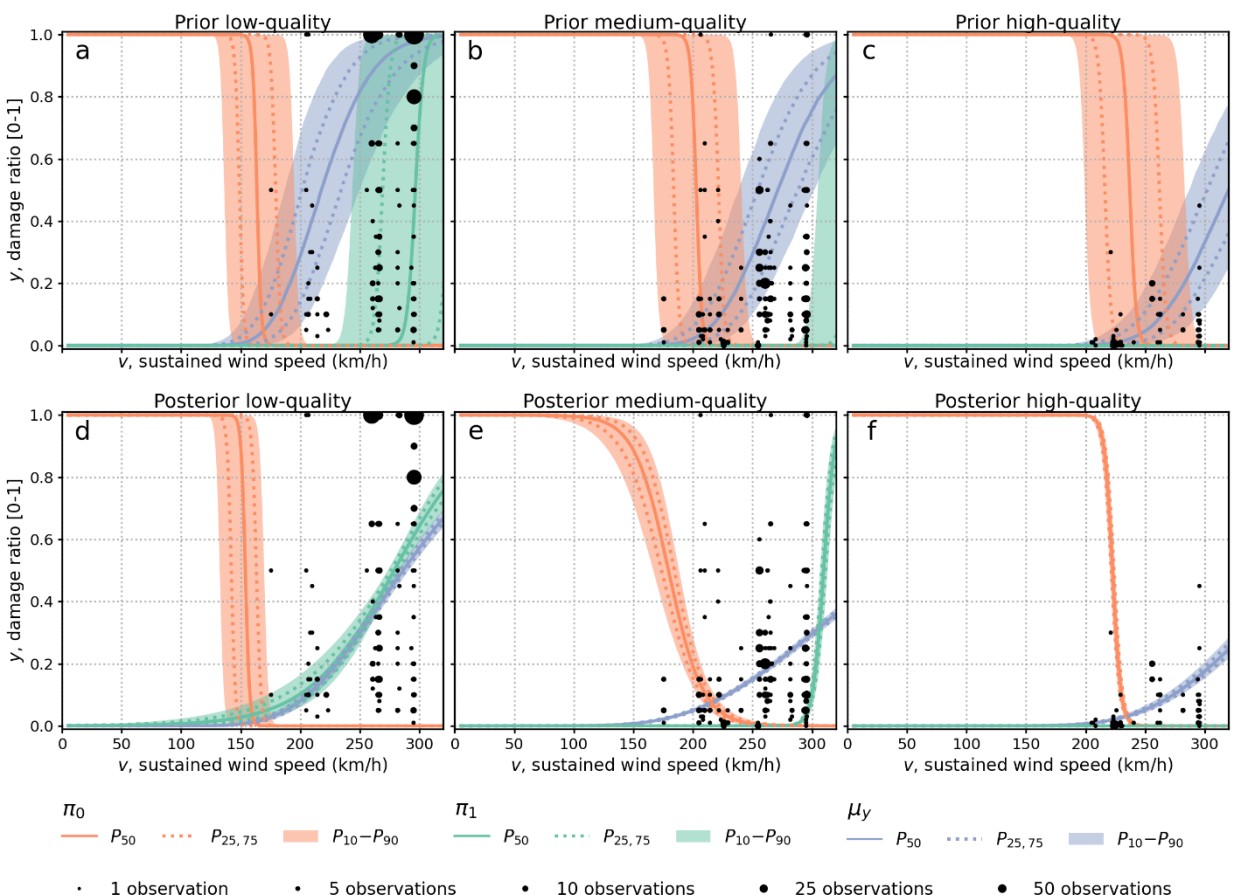

**Figure 5: Visualization of observations, priors and posteriors 25th, 50th (median), 75th percentile and the 10th–90th percentile range for low-, medium- and high-quality buildings.**

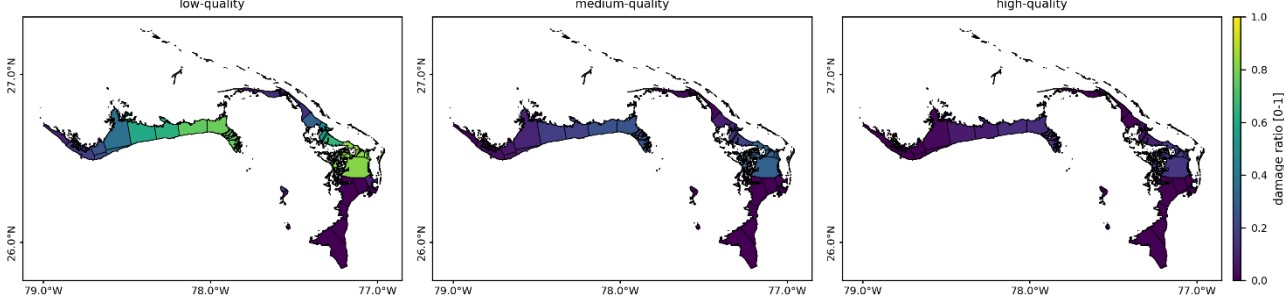

**Figure 6: Posterior damage ratio per district for low-, medium- and high-quality buildings.**

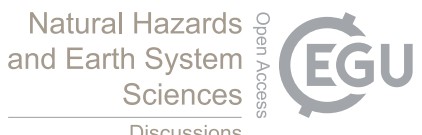

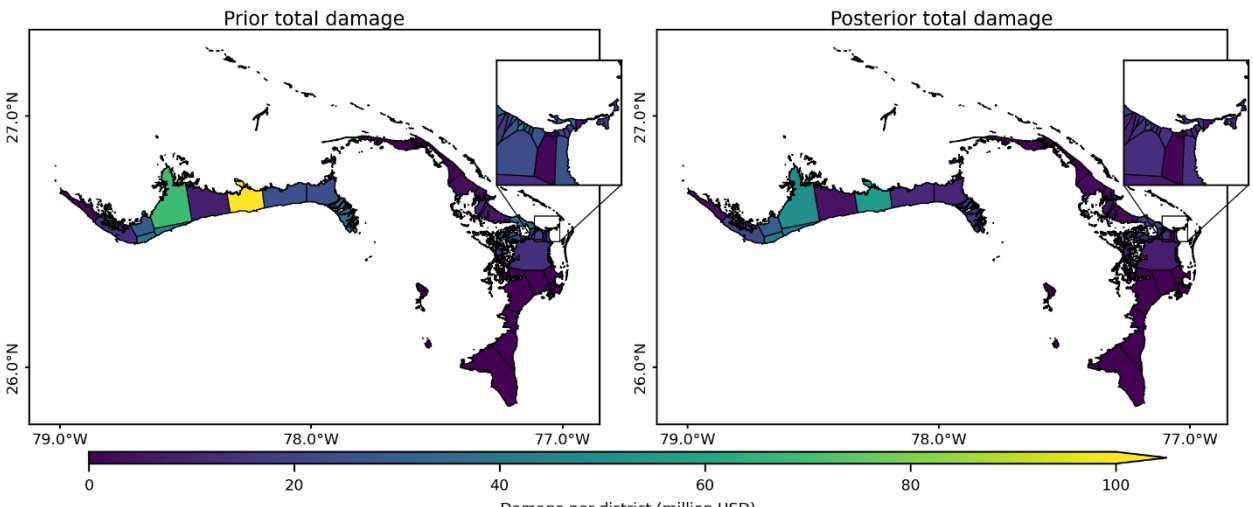

**Figure 7: Damage to low-, medium- and high-quality housing combined derived from prior and posterior vulnerability curves.**

**Table 1: Damage for residential buildings per building class derived from prior and posterior distributions**

|  | **Prior (million USD)** | **Posterior (million USD)** | **Percentage change** |
|---|---|---|---|
| *Low-quality* | 155 | 124 | -20% |
| *Medium-quality* | 543 | 324 | -40% |
| *High-quality* | 359 | 210 | -41% |
| ***Total*** | **1056** | **658** | **-38%** |