# Peer review of "Using rapid damage observations from social media for Bayesian updating of hurricane vulnerability functions: A case study of Hurricane Dorian"

_Natural Hazards and Earth System Sciences, 2020_

## Referee Comment (RC1) · Anonymous Referee #1 · 26 Oct 2020

The authors present an interesting topic for quantifying losses from a hurricane. While the paper has a lot of potential, I think the following comments will ensure that the paper has merit across hazard types and research fields.

The paper builds on important work performed on estimating damages from imagery. However, the current literature review and methodology sections are cursory at best and lack significant details for using images in a damage assessment. For instance, the literature review is missing any detail on quantifying structural damages through survey such as using the Tornado Injury Scale (TIS; Curtis & Fagan, 2013), or papers

such as Meyer and Hendricks (2018) which directly measure damages and recovery using images. Like the literature review, the methods section was limited in terms of image estimates. The authors do a great job outlining the model they use for the final analysis, but their contribution is using youtube to inform those models. The authors allocate two pages to explaining the Bayesian model, but they spend two paragraphs explaining how they collected the imagery. This lack of detail limits the use of this paper for other researchers and removes any valid reproducibility.

Several questions I asked while reading the methods included:1) how many people watched the videos and quantified damages? 2) If more than one person was watching the videos and estimating damages how did the authors deal with potential issues with inter-rater reliability (See meyer and Hendricks for example)? 3) How did you rate damages? Without answers to questions like these the generalizability of the study is severely limited.

Within the results section I was disappointed to not see a section on how these measures were validated. I don't feel it is enough to say "total damages are lower with this new model", without first giving evidence as to how your estimates improved the calculations. Without these validation metrics from a test dataset, the findings can't be assumed to improve the model and may in fact be making it worse.

Meyer, Michelle Annette, and Marccus D. Hendricks. "Using photography to assess housing damage and rebuilding progress for disaster recovery planning." Journal of the American Planning Association 84.2 (2018): 127-144. Curtis , A. , & Fagan , W. F. ( 2013 ). Capturing damage assessment with a spatial video: An example of a building and street-scale analysis of tornado-related mortality in Joplin, Missouri, 2011 . Annals of the Association of American Geographers , 103 ( 6 ), 1522 – 1538 . doi: 10.1080/00045608.2013.784098

---

## Referee Comment (RC2) · Anonymous Referee #2 · 14 Nov 2020

The manuscript at hand presents a Bayesian approach to the updating of vulnerability functions for the rapid forecasting of natural catastrophe damage. Based on social media, the authors demonstrate the adaptation of generic vulnerability functions to Hurricane Dorian. The manuscript is well written and describes the novel approach in great clarity. The presented framework is highly relevant, in particular for practitioners in the field. With a strong deviation in terms of damage estimates, the results of the analysis provide a good argument for further investigation into the adaptation of vulnerability curves.

[Figure]

Unfortunately, the manuscript lacks a deeper analysis of the obtained results and does not create further scientific insight beyond the presentation of the framework. The approach could have been easily applied to a historic event for which empirical damage estimates were already available. These could have served as a benchmark for damage estimates from both the generic and the adapted vulnerability curves, providing evidence for the otherwise hypothetical improvement.

While I tend to follow the author's claim that the proposed Bayesian framework will deliver improved damage estimates, some factual evidence should be given. This could be either in the form of quantitative validation or comparison with comparable vulnerability curves. Given the likelihood of substantial bias in social media accounts, the author's should demonstrate that the updated vulnerability functions are in fact closing and not unintentionally widening the gap between model and reality.

**Further comments:**

Lines 70 ff.: The authors state that they aim to improve existing vulnerability functions. However they fail to produce evidence of this claim. In lines 215 ff. the authors explain that the calibration of the prior is based on expert judgement. Due to lack of reference, the reader does neither know how accurate damage estimates based on the prior, nor how accurate those based on the posterior vulnerability curves are.

A key feature for the proposed framework is the use of the zero-one inflated beta distribution. Yet, the results cover only the mean of the beta distribution, no results for $\phi$ are given. A posterior distribution should be given. It would be interesting to see a plot of uncertainty intervals around $\mu_y$ based on the precision of the beta distribution. The authors could discuss how this information could be leveraged to provide meaningful uncertainty bands for regional damage estimates.

Lines 263 ff.: The authors argue that adherence to building codes is one reason why the posterior vulnerability curves for medium- and high-quality building are considerably lower than the prior. But shouldn't it be the key assumption for the prior belief that

buildings conform to publicly known building codes? Doesn't it first of all suggest that the prior belief was too high?

Appendix D: It seems impossible to reconcile the values shown in plots 1-3 for $\Theta_1, ..., \Theta_6$ with either the prior or posterior curves. According to Figure 5 the median capacity $\Theta_1$ (equivalent to $\alpha$ in Eq. 2) should be somewhere between 200 and 250 km/h for low-quality buildings. However D1 gives a mean of roughly 80-90 (unit?). The authors mention that for Gibbs sampling $v$ and all inputs were re-scaled by the maximum observed velocity, but in this case, one would expect $0 < \Theta_1 < 1$. It is similarly impossible to reconcile the parameters of the logistic curves $\pi_0$ and $\pi_1$, where a probability of 0.5 should be reached at $v_{1/2} = -\Theta_3/\Theta_4$ or $v_{1/2} = -\Theta_5/\Theta_6$, respectively. Values for $\Theta_1, ...\Theta_6$ should be corresponding to the the results shown and be given on an interpretable scale.

**Minor comment**:

Lines 56 ff.: The authors write that the Bayesian approach is an example for methods to improve vulnerability curves from observations. What other methods?

Figure D3: Figure is barely readable due to very poor image resolution. Please provide publication-grade quality

Lines 323 ff.: Please explain which mean you refer when setting the standard deviations. Presumably $\mu_\Theta$.

Typo in line 264: 'designed to be designed'

---

## Author Comment (AC1) · 25 Jan 2021

**The paper builds on important work performed on estimating damages from imagery. However, the current literature review and methodology sections are cursory at best and lack significant details for using images in a damage assessment. For instance, the literature review is missing any detail on quantifying structural damages through survey such as using the Tornado Injury Scale (TIS; Curtis & Fagan, 2013), or papers such as Meyer and Hendricks (2018) which directly measure damages and recovery using images. Like the literature review,**

**the methods section was limited in terms of image estimates. The authors do a great job outlining the model they use for the final analysis, but their contribution is using youtube to inform those models. The authors allocate two pages to explaining the Bayesian model, but they spend two paragraphs explaining how they collected the imagery. This lack of detail limits the use of this paper for other researchers and removes any valid reproducibility.**

First of all, we thank the reviewer for your kind words and suggestions. Below, I will list the reviewers points in order and discuss their suggestions. Here, we also specify specifically how we aim to make the contribution more reproducible.

**Several questions I asked while reading the methods included:1) how many people watched the videos and quantified damages? 2) If more than one person was watching the videos and estimating damages how did the authors deal with potential issues with inter-rater reliability (See meyer and Hendricks for example)? 3) How did you rate damages? Without answers to questions like these the generalizability of the study is severely limited.**

Thank you for your suggestion. We will in more detail discuss literature on quantifying structural damages through images, for example by including a map of the observations and their source (i.e., observation from ground or air).

Unfortunately, the scales suggested by Curtis & Fagan (2013) and Meyer and Hendricks (2018) use a damage score (TIS1-10 and 0-9 respectively), but do not translate these scores into a damage ratio.

However, we fully agree with the reviewer that this leads to problems regarding reproducibility, especially because the damage was assessed by only one person (James E. Daniell) leading to subjectivity in the scores. In fact, this is a common problem when human judges are used. A solution to this problem is to use multiple judges assuming that the average or median assessment of multiple judges will lead to a more objective, and reproducible, score.

Therefore, to obtain a more objective score Antonios Pomonis and Joshua Macabuag assessed the damages and building classes in each image as well resulting in three damage classifications for each image by engineers experienced in assessing building damages after disasters. Then, following Meyer and Hendricks (2018) and others we calculated intercoder reliability tests. Subsequently, we will use the median damage ratio and building class for further analysis.

From a preliminary analysis of the scores for the individual buildings with three judges, where each judge rates each target, we obtain an intraclass correlation of 0.92 for building damages using the Spearman Brown adjusted reliability and a Fleiss kappa of 0.30 for building class.

**Within the results section I was disappointed to not see a section on how these measures were validated. I don't feel it is enough to say "total damages are lower with this new model", without first giving evidence as to how your estimates improved the calculations. Without these validation metrics from a test dataset, the findings can't be assumed to improve the model and may in fact be making it worse.**

We fully agree with the reviewer that a comparison with gold standard vulnerability curves would be beneficial. A (cursory) damage report is available for hurricane Dorian (ECLAC et al., 2019). However, this damage report only reports total damages. This means that 1) the report also includes damage from the storm surge, and 2) no vulnerability curves are presented hindering a direct comparison of vulnerability curves.

To make a true comparison, between our vulnerability curve and a gold standard vulnerability curve, we would require not just information about risk (the damages), but rather about the vulnerability component of risk. However, here we run into two problems:

1. To the best of our knowledge no independent wind vulnerability curves are available for hurricane Dorian in the Bahamas, while comparison with similar events in similar locations would neglect the purpose of this manuscript of creating event-specific vulnerability curves.

2. Even if vulnerability curves were available, these are dependent on both the hazard component and damage observations. Since wind speeds as part of the uncertainty within the process, many vulnerability functions being characterized as semi-empirical (e.g., Mason and Parackal, 2015; Pita et al., 2015; Smith et al., 2020; Walker, 2011) and direct comparison would be unproductive.

Moreover, other vulnerability curves and observations would also be prone to uncertainties. Therefore, we believe that by updating previously existing evidence with new data, we are in fact converging towards the true vulnerability curve. Any additional observations should be treated as additional evidence rather than test data.

However, we should make this clearer in the manuscript and if we were allowed to submit a revised version of our manuscript, we shall include a discussion to this extent. In addition, we will revise several sentences. For example, *"total damages are lower with this new model"* as quoted by the reviewer to *"using the posterior vulnerability curves total damages are projected to be lower"*.

**References**

Curtis, A. and Fagan, W. F.: Capturing damage assessment with a spatial video: An example of a building and street-scale analysis of tornado-related mortality in Joplin, Missouri, 2011, Ann. Assoc. Am. Geogr., 103(6), 1522–1538, 2013.

ECLAC, IDB, PAHO and WHO: Assessment of the Effects and Impacts of Hurricane Dorian in the Bahamas., 2019.

Mason, M. S. and Parackal, K. I.: Vulnerability of buildings and civil infrastructure to tropical cyclones: a preliminary review of modelling approaches and literature, Bush.

Nat. Hazards CRC, Melb., 2015.

Meyer, M. A. and Hendricks, M. D.: Using photography to assess housing damage and rebuilding progress for disaster recovery planning, J. Am. Plan. Assoc., 84(2), 127–144, 2018.

Pita, G., Pinelli, J. P., Gurley, K. and Mitrani-Reiser, J.: State of the art of hurricane vulnerability estimation methods: A review, Nat. Hazards Rev., 16(2), 1–16, doi:10.1061/(ASCE)NH.1527-6996.0000153, 2015.

Smith, D. J., Edwards, M., Parackal, K., Ginger, J., Henderson, D., Ryu, H. and Wehner, M.: Modelling vulnerability of Australian housing to severe wind events: past and present, Aust. J. Struct. Eng., 21(3), 175–192, 2020.

Walker, G. R.: Modelling the vulnerability of buildings to wind—a review, Can. J. Civ. Eng., 38(9), 1031–1039, 2011.

---

## Author Comment (AC2) · 25 Jan 2021

**The manuscript at hand presents a Bayesian approach to the updating of vulnerability functions for the rapid forecasting of natural catastrophe damage. Based on social media, the authors demonstrate the adaptation of generic vulnerability functions to Hurricane Dorian. The manuscript is well written and describes the novel approach in great clarity. The presented framework is highly relevant, in particular for practitioners in the field. With a strong deviation in terms of damage estimates, the results of the analysis provide a good argument for further**

[Figure]

**investigation into the adaptation of vulnerability curves**.

We thank the reviewer for their kind words. Below, I will list the reviewers points in order and discuss their suggestions.

**Unfortunately, the manuscript lacks a deeper analysis of the obtained results and does not create further scientific insight beyond the presentation of the framework. The approach could have been easily applied to a historic event for which empirical damage estimates were already available. These could have served as a benchmark for damage estimates from both the generic and the adapted vulnerability curves, providing evidence for the otherwise hypothetical improvement.**

We fully agree with the reviewer that a comparison with gold standard damage estimates would be beneficial. A (cursory) damage report is available for hurricane Dorian (ECLAC et al., 2019). However, this damage report only reports total damages. This means that 1) the report also includes damage from the storm surge, and 2) no vulnerability curves are presented hindering a direct comparison of vulnerability curves.

For some historic events more detailed damage are available. However, to make a true comparison, between our vulnerability curve and a gold standard vulnerability curve, we would require not just information about risk (the damages), but rather about the vulnerability component of risk. However, here we run into two problems:

1. To the best of our knowledge no independent wind vulnerability curves are available for hurricane Dorian in the Bahamas, while comparison with similar events in similar locations would neglect the purpose of this manuscript of creating event-specific vulnerability curves.

2. Even if vulnerability curves were available, these are dependent on both the hazard component and damage observations. Since wind speeds as part of the uncertainty within the process, many vulnerability functions being characterized

as semi-empirical (e.g., Mason and Parackal, 2015; Pita et al., 2015; Smith et al., 2020; Walker, 2011) and direct comparison would be unproductive.

Moreover, other vulnerability curves and observations would also be prone to uncertainties. Therefore, we believe that by updating previously existing evidence with new data, we are in fact converging towards the true vulnerability curve. Any additional observations should be treated as additional evidence rather than test data.

However, we should make this clearer in the manuscript and if we were allowed to submit a revised version of our manuscript, we shall include a statement to this extent. In addition, we will revise several sentences. For example, *"total damages are lower with this new model"* to *"using the posterior vulnerability curves total damages are projected to be lower"*.

**While I tend to follow the author's claim that the proposed Bayesian framework will deliver improved damage estimates, some factual evidence should be given. This could be either in the form of quantitative validation or comparison with comparable vulnerability curves. Given the likelihood of substantial bias in social media accounts, the author's should demonstrate that the updated vulnerability functions are in fact closing and not unintentionally widening the gap between model and reality.**

At the same time, we follow the reviewer in the potential biases in social media. We identify 2 main biases:

1. *A judgement bias* due to the damage ratios estimated by a single judge.

2. An *availability bias* because social media data is likely to focus more on heavily impacted areas.

To reduce the effect of the *judgement bias*, we will use multiple judges in an updated manuscript. Therefore, to obtain a more objective score Antonios Pomonis and Joshua Macabuag assessed the damages and building classes in each image in addition to James Daniell resulting in three damage classifications for each image by engineers experienced in assessing building damages after disasters. Then, following Meyer and Hendricks (2018) and others we calculated intercoder reliability tests. Subsequently, will use the median damage ratio and building class for further analysis.

From a preliminary analysis of the scores for the individual buildings with three judges, where each judge rates each target, we obtain an intraclass correlation of 0.92 for building damages using the Spearman Brown adjusted reliability and a Fleiss kappa of 0.30 for building class.

To reduce the effect of the *availability bias* we have only used videos that show an overview of an area, and most cases, flyovers of entire islands, rather than just the most impacted buildings (l. 200).

**Lines 70 ff.: The authors state that they aim to improve existing vulnerability functions. However they fail to produce evidence of this claim. In lines 215 ff. the authors explain that the calibration of the prior is based on expert judgement. Due to lack of reference, the reader does neither know how accurate damage estimates based on the prior, nor how accurate those based on the posterior vulnerability curves are.**

See discussion above about the validation of vulnerability curves.

**A key feature for the proposed framework is the use of the zero-one inflated beta distribution. Yet, the results cover only the mean of the beta distribution, no results for $\phi$ are given. A posterior distribution should be given. It would be interesting to see a plot of uncertainty intervals around y based on the precision of the beta distribution. The authors could discuss how this information could be leveraged to provide meaningful uncertainty bands for regional damage esti-**

**mates.**

We will provide results for $\phi$, see figure below.

**Lines 263 ff.: The authors argue that adherence to building codes is one reason why the posterior vulnerability curves for medium- and high-quality building are considerably lower than the prior. But shouldn't it be the key assumption for the prior belief that buildings conform to publicly known building codes? Doesn't it first of all suggest that the prior belief was too high?**

The vulnerability curves are based on our best available estimate of the vulnerability curve at the start of the damage assessment, which was then updated using social media data. Given the additional data available we updated the hypothesis and now belief that this is the reason our initial judgement was too low because more buildings were built according to those building codes than initially thought. The building codes themselves were publicly available, the number of buildings that were built using these building codes was unknown. However, since this is solely based on expert judgement, another expert might have judged differently – precisely why we believe why we should update prior beliefs with any available additional evidence. Therefore, we removed this sentence from the manuscript, which now states: *"The largest relative differences were found for medium- and high-quality buildings".*

**Appendix D: It seems impossible to reconcile the values shown in plots 1-3 for $\theta_1, \ldots, \theta_6$ with either the prior or posterior curves. According to Figure 5 the median capacity $\theta_1$ (equivalent to in Eq. 2) should be somewhere between 200 and 250 km/h for low-quality buildings. However D1 gives a mean of roughly 80-90 (unit?). The authors mention that for Gibbs sampling $v$ and all inputs were re-scaled by the maximum observed velocity, but in this case, one would expect $0 < \theta_1 < 1$. It is similarly impossible to reconcile the parameters of the logistic curves $\pi_0$ and $\pi_1$, where a probability of 0.5 should be reached at $v_{\frac{1}{2}} = -\theta_1/\theta_4$ or $v_{\frac{1}{2}} = -\theta_5/\theta_6$, respectively. Values for $\theta_1, \ldots, \theta_6$ should be corresponding to the**

**the results shown and be given on an interpretable scale.**

Here, we thank the reviewer for their good eye. The plots were mixed up and should make sense in the revised version (see also figure above). In addition, we will include a notice in the caption that the values shown in the plots are denormalized.

**Lines 56 ff.: The authors write that the Bayesian approach is an example for methods to improve vulnerability curves from observations. What other methods?**

This sentence was indeed phrased incorrectly and is now updated to "A scientific challenge is to seek for methods that use observations from the affected area to improve vulnerability curves. A method that can employ observational data to update prior beliefs (e.g., beliefs based on expert judgment), is Bayesian analysis"

**Figure D3: Figure is barely readable due to very poor image resolution. Please provide publication-grade quality**

We provided a higher quality image (see figure below). In addition, the figure is now transposed allowing more space for the graphs in portrait mode.

**Lines 323 ff.: Please explain which mean you refer when setting the standard deviations. Presumably .**

 We will include the mean to which we refer. This is indeed $\mu_\theta$.

**Typo in line 264: 'designed to be designed'**

 We corrected this typo.

**References**

ECLAC, IDB, PAHO and WHO: Assessment of the Effects and Impacts of Hurricane Dorian in the Bahamas., 2019.

Mason, M. S. and Parackal, K. I.: Vulnerability of buildings and civil infrastructure to tropical cyclones: a preliminary review of modelling approaches and literature, Bush. Nat. Hazards CRC, Melb., 2015.

Meyer, M. A. and Hendricks, M. D.: Using photography to assess housing damage and rebuilding progress for disaster recovery planning, J. Am. Plan. Assoc., 84(2), 127–144, 2018.

Pita, G., Pinelli, J. P., Gurley, K. and Mitrani-Reiser, J.: State of the art of hurricane vulnerability estimation methods: A review, Nat. Hazards Rev., 16(2), 1–16, doi:10.1061/(ASCE)NH.1527-6996.0000153, 2015.

Smith, D. J., Edwards, M., Parackal, K., Ginger, J., Henderson, D., Ryu, H. and Wehner, M.: Modelling vulnerability of Australian housing to severe wind events: past and present, Aust. J. Struct. Eng., 21(3), 175–192, 2020.

Walker, G. R.: Modelling the vulnerability of buildings to wind—a review, Can. J. Civ. Eng., 38(9), 1031–1039, 2011.
* * *
[Figure]

Fig. 1.